# GIR Framework: Learning Graph Positional Embeddings with Anchor Indication and Path Encoding

## Abstract

The majority of existing graph neural networks (GNNs) following the message passing neural network (MPNN) pattern have limited power in capturing position information for a given node. To solve such problems, recent works exploit positioning nodes with selected anchors or the message passing destination, mostly in a process that first explicitly assign distances information and then perform message passing encoding. However, those existing two-stage models ignore potentially useful interaction between intermediate results of the distance computing and encoding stages to some extent, due to the model architecture designs. In this work, we propose a novel framework which follows the anchor-based idea and aims at conveying distance information implicitly along the MPNN message passing steps for encoding position information, node attributes, and graph structure in a more flexible way. Specifically, we first leverage a simple anchor indication strategy to enable the position-aware ability for well-designed MPNNs. Then, following this strategy, we propose the Graph Inference Representation (GIR) model, which acts as a generalization of MPNNs with a more specific propagation path design for position-aware scenarios. Meanwhile, we theoretically and empirically explore the ability of the proposed framework to get position-aware embeddings, and experimental results show that our proposed method generally outperforms previous position-aware GNN methods.

## 1 Introduction

Graph, as an important data structure, is a powerful tool to represent ubiquitous relationships in the real world. Learning vector representations for graph data, benefits many downstream tasks on the graph such as node classification (Kipf & Welling, 2017) and link prediction (Zhang & Chen, 2018). Many graph representation learning methods have been proposed recently, among those, Graph Neural Networks (GNNs), inheriting the merits of neural networks, have shown superior performance and become a much popular choice.

Existing GNN models mainly follow the message passing neural network (MPNN) (Gilmer et al., 2017) pattern, which stacks message passing layers that aggregate information from neighborhoods and then update representations for each node. Typical MPNNs are limited by 1-Weisfeiler-Lehman test (Xu et al., 2019), and lack of ability to capture the position information within the graph (You et al., 2019), without distinguishable node/edge attributes, nodes in a different part of the graph with topologically equivalent neighborhood structures or even with different substructures may be embedded into the identical representation by typical MPNNs alone (You et al., 2019; Li et al., 2020), as shown in figure 1 (b), $A_1$ and $B_2$ cannot be distinguished with MPNNs and no distinguishable attributes (when ignoring colored anchor nodes).

Researchers have developed methods to alleviate this issue. Some earlier works adopt one-hot encodings as extended node attributes (Kipf & Welling, 2017). More recent methods utilize graph distance information to get position-aware embeddings. Anchor-based GNNs (You et al., 2019; Liu et al., 2019) select anchor nodes as positioning bases, and use position information related to anchors to break the structural symmetry (Figure 1 (c)). Distance encoding (Li et al., 2020) utilizes distance information to push typical MPNNs beyond the 1-Weisfeiler-Lehman test limitation (Figure 1 d).

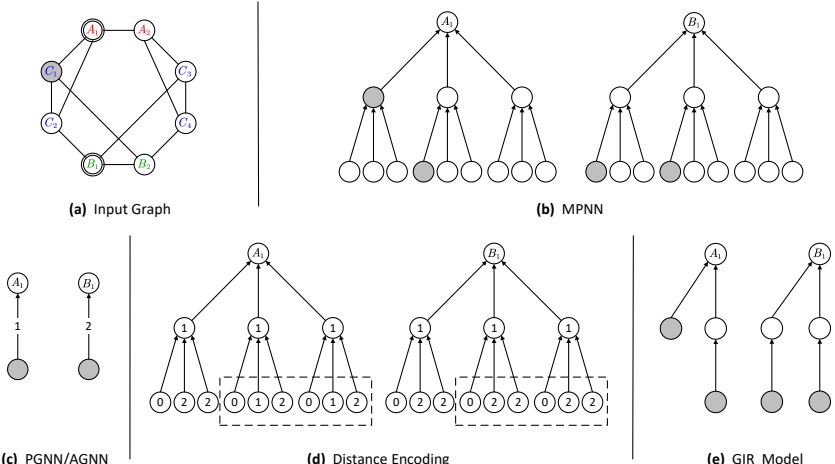

Figure 1: Comparison of position-aware GNN methods. **(a)** (Modified from (Li et al., 2020))Example unattributed input graph, nodes start with the same letter ($A$, $B$ or $C$) are structurally equivalent. Nodes in a doubled circle ($A_1$ and $B_1$) are selected for demonstrating different position-aware GNN methods; the colored node ($C_1$) will be chosen as an anchor for anchor-based GNNs. **(b-e)** Different position aware GNN methods: MPNN (with anchor labeling); PGNN/AGNN; Distance Encoding; GIR model.

Those existing position-aware GNN methods mainly explicitly pre-compute the distance between node pairs, utilizing them as node attributes or controlling of message passing steps, and then perform message passing encoding. Due to the design choices for what and how to utilize distance information, those existing methods consider graph structure and attributes information encoded by GNN propagation and position information in a relatively separate manner, potentially useful interaction between intermediate results of the distance computing and encoding stages may be ignored.

In this paper, we follow the anchor-based GNN strategy and aim at performing a more flexible interaction with graph structure, positioning and attribute information. With limited fixed anchor positioning base selected, abundant position information could be assigned, boosting the performance of MPNNs in the position aware scenarios in a trackable way. Besides the explicit assigning strategy, here we mainly focus on conveying position information implicitly, in order to retain the potential possibility for more flexible usage of the graph data, as some information beyond pre-performed graph algorithms (e.g. shortest path distance algorithm) may potentially useful to be captured. Our motivation comes from the structural similarity between GNN message passing and distance relaxation process of the Bellman-Ford shortest path algorithm (Bellman, 1958), we show that with a simple indication of anchors and appropriate design for message passing functions, MPNNs can keep track of the shortest paths from anchors. Further, inspired by a relaxation order improved variant of Bellman-Ford algorithm (Moore, 1959), we propose a generalized MPNN architecture, termed Graph Inference Representation (GIR), as a more specialized model for the position-aware scenario. The GIR model propagates messages from anchors along paths to each node, and outputs of the $k$-th layer encode $k$-step representations related to anchors. On an unweighted graph with no distinguishable node attributes, the proposed strategies provide natural ways to help break the structural symmetry as the previous anchor-based GNNs could (Figure 1 (b&e)).

Our contributions are summarized as follows:

1. We propose a general anchor labeling strategy, enabling MPNNs for mimicking the Bellman-Ford algorithm and getting position-aware embeddings implicitly under anchor-based GNN pattern and discuss theoretical implications and experimental realizations.

2. We propose the Graph Inference Representation (GIR) model that contains more specialized structure for learning position-aware embeddings.

3. Empirically, we evaluate the performance of proposed methods for tasks in position-aware datasets, experimental results show that our position-aware GIRs achieve generally higher performance.

The rest of the paper is organized as follows. Section 2 reviews related works. Section 3 introduces notations and definitions, theoretical implications are discussed. Section 4 details the proposed GIR framework. Section 5 presents the experiment and gives a discussion of the results. Finally, section 6 presents our conclusion.

## 2 RELATED WORKS

Our work follows the anchor-based GNN pattern from the position-aware GNN literature, with the design inspired by the Bellman-Ford algorithm.

### 2.1 POSITION-AWARE GNNS

Position-aware GNNs leverage distance information in the message propagating encoding process. Distance Encoding (Li et al., 2020) follows the MPNN propagation design and extends it with pre-assigned graph distance from the propagation target to each node (Figure 1, (d)). For each message propagation step on a target node $v$, only distance from $v$ are utilized, potentially useful distance from other nodes are not considered.

Anchor-based GNNs use selected anchor nodes as a positioning base, existing methods mainly follow the two-stage pattern: select anchor nodes first, and then encode the information related to anchors. Position-aware GNN (PGNN) (You et al., 2019) selects anchor node sets randomly before running every forward of the model to get a low distortion embedding capturing global position information, a PGNN layer directly propagates message from the selected anchor node set to each target node, weighted by the pairwise distance. The random anchor selecting strategy of PGNNs leads to unstable limitation, some more recent work uses fixed anchor nodes instead to overcome this issue. AGNN (Liu et al., 2019) pre-selects fixed anchors by minimum point cover nodes algorithm; GraphReach (Nishad et al., 2020) follows the fix anchor setting and adopts random walk reachability estimations instead of the shortest path distance. Those methods adopt the strategy that ignores graph structure when performing message propagation.

Our GIR framework follows the anchor-based GNN pattern and differs from previous works in the position information encoding and utilizing strategy (depicted in figure 1). GIRs are capable of encoding position information implicitly, taken intermediate results in computing shortest path distance into consideration, and under specific settings, corresponding distances related to anchors and the graph structure are utilized together.

### 2.2 BELLMAN-FORD ALGORITHM

The Bellman-Ford algorithm (Bellman, 1958) computes the shortest path distance from a single source node to each node in a weighted digraph. For a graph $\mathcal{G} = (\mathbb{V}, \mathcal{E})$, source node $s$, the algorithm performs $|\mathbb{V}| - 1$ relaxation iterations, maintaining the shortest path from $s$ with at most $i$ edges in the $i$-th iteration. A relaxation step iterate over all edges, correct distance to better ones. The relaxation of distance on $i$-th iteration and edge $\langle u, v \rangle$ with weight $w$ is defined as,

$$dist_v^i = min(dist_v^{i-1}, dist_u^{i-1} + w) \quad i > 0 \tag{1}$$

where $dist^0$ is initialized as,

$$dist_v^0 = \begin{cases} 0 & v = s \\ \infty & otherwise \end{cases} \tag{2}$$

A variant of Bellman-Ford algorithm (Moore, 1959) notices that if the distance value of node $v$ has not been changed since its last relaxed, edges out of $v$ are no need to perform relaxation.

Learning graph algorithm including the Bellman-Ford algorithm with MPNN has been experimented in Velikovi et al. (2020), they show the effectiveness of adopting max-pooling as aggregator, and propose to learn with intermediate results of graph algorithms. Our work goes further toward the implication from the Bellman-Ford algorithm, and proposes a more specialized message passing strategy beyond typical MPNN for this. In addition, experiment setting of Velikovi et al. (2020) focus on a single specific source node, and in the experiment on synthetic datasets, we generalize this to adapt the anchor-based GNN settings.

Table 1: Notations

| Notation | Description |
|---|---|
| $\mathcal{G}$ | the input graph |
| $\mathbb{V}, \mathcal{E}$ | the node/edge set of $\mathcal{G}$ |
| $v_i$ | the i-th node in $\mathcal{G}$ |
| $X, Z$ | the node attributes/representations of $\mathcal{G}$ |
| $x_i, z_i$ | the attribute/representation of $v_i$ |
| $W$ | the edge weights of $\mathcal{G}$ |
| $\mathcal{N}(v)$ | in-neighborhoods of node $v$ in $\mathcal{G}$ |
| $\mathbb{A}$ | the anchor node set |
| $Successors(\mathcal{G}, \mathbb{S})$ | successors of any node in source node set $\mathbb{S} \subseteq \mathbb{V}$ |
| $[\boldsymbol{a}||\boldsymbol{b}]$ | concat vector $\boldsymbol{a}$ and $\boldsymbol{b}$ |

## 3 PRELIMINARIES

### 3.1 NOTATIONS

A graph can be represented as $\mathcal{G} = (\mathbb{V}, \mathcal{E})$, where $\mathbb{V} = \{v_1, \cdots, v_n\}$ is the node set and $\mathcal{E} = \{\langle v_i, v_j \rangle | v_i, v_j \in \mathbb{V}\}$ is the edge set. Nodes are augmented with the feature matrix $X$, which is either input attributes if available or placeholders. Edges may augmented with the optional weights $W$. In-neighborhoods of node $v$ are represented as $\mathcal{N}(v)$. Notations are summarized in table 1.

### 3.2 POSITION-AWARE EMBEDDINGS

One goal of the anchor-based GNN model is to utilize anchors as bases to encode position-aware information for each node. To capture this intuition, PGNNs (You et al., 2019) view embeddings as position-aware if the shortest path distance between node pairs could be reconstructed from their embeddings, which is hard for models with fixed anchors to satisfy. Here we define position-aware embeddings related to anchors (Definition 1). Considering that the shorest path distance to a node set has different definitions, such as the shortest path distance to all anchor nodes or to any node in the anchor node set, or whether to focus on shorest path with limited hop. Here in the definition 1, we focus on the limited hop shortest path distance (for the alignment with MPNN) to any node in specific node sets, for retaining a high generality, and the union of those node sets need to be the anchor node set, for consisting with the position-aware embeddings related to anchors definition.

**Definition 1** (Position-aware Embeddings). *For $\mathscr{A} \subseteq \mathcal{P}(\mathbb{A}), \bigcup_{\mathscr{A}} = \mathbb{A}$, where $\mathcal{P}(\mathbb{A})$ is the power set of node set $\mathbb{A}$, the node embedding $Z = \{z_i = f(v_i) | \forall v_i \in \mathbb{V}\}$, where $f$ is a graph encoder that maps a node to its embedding, is k-hop $\mathscr{A}$-position-aware if there exists functions $\{g_{\mathbb{A}'}(\cdot) | \forall \mathbb{A}' \in \mathscr{A}\}$ such that $g_{\mathbb{A}'}(z_i) = d_{sp}(v_i, \mathbb{A}')$, where $d_{sp}(v_i, \mathbb{A}')$ is the k-hop shortest path distance between node $v_i$ and any node in the node set $\mathbb{A}'$ in the graph $\mathcal{G}$. If all elements in $\mathscr{A}$ are singleton, brakets of sets inner $\mathscr{A}$ could be omitted.*

As examples for definition 1, we could reconstruct $k$-hop shortest path distance from any node in the anchor set $\mathbb{A}$ from $k$-hop $\{\mathbb{A}\}$-position-aware embeddings, and $k$-hop shortest path distances from all anchor nodes from $k$-hop $\mathbb{A}$-position-aware embeddings, these two settings will be discussed most commonly in following sections.

### 3.3 INDICATABILITY

Here we discuss the theoretical implicitly of our GIR framework design. We begin with our motivation of mimicking the Bellman-Ford algorithm with MPNNs, the concept of indicatability is introduced as an effective tool for the design of the GIR framework.

A natural idea for an MPNN to mimic the Bellman-Ford algorithm is to perform neural relaxation functions and enable the capability of keeping track of the intermediate limited hop shortest path distances. The motivation of mimicking the Bellman-Ford algorithm and getting position-aware embeddings implies the capability for reconstructing $k$-hop shortest path distance from $k$-th layer.

To capture this intuition, we introduce the concept of **indicatability** that implies the existence and constructibility of functions that reconstruct specific information related to a node set from a vector representation.

**Definition 2** (Indicatability). *For $\mathbb{A}' \subseteq \mathbb{A}$, the function set $f$ is $\mathbb{A}'$-indicatable over function $f$ if there exists a constructable function $f_{\mathbb{A}'}$ such that for all $v_i \in \mathbb{A}$, $f_{\mathbb{A}'}(z_i) = f(v_i, \mathbb{A}')$.*

The indicatability (Definition 2) is defined over a pre-defined function $f$ that maps a node $v_i$ and a node set $\mathbb{A}'$ to a real value (e.g. the multi-souce shortest path distance function), which could be mimicked by a neural network module that takes $z_i$, the embedding of $v_i$ as input and could generate outputs include $f(z_i, \mathbb{A}')$, as the target neural network module $f$ is capable for mimicking $f$ over $\mathbb{A}'$, we say that $f$ is $\mathbb{A}'$-indicatable over $f$. Moreover, the $k$-hop $\mathscr{A}$-position-aware definition (Definition 1) can be restated as the existence of f-dist with $\mathbb{A}'$-indicatability over $k$-hop limited multi-souce shortest path distance function, for all $\mathbb{A}' \in \mathscr{A}$.

### 3.4 Neural Bellman-Ford with MPNNs

Here we show that a well-designed and parameterized MPNN is capable for representing the process of updating the existence of an $\mathbb{A}'$-indicatable f-dist$^{k-1}$ for $Z^{k-1}$ to $\mathbb{A}'$-indicatable f-dist$^k$ over $Z^k$ in the $k$-th layer, the mentioned corresponding functions are directly implemented or implicitly mimicked by well-learned parameterized multi-layer perceptrons (MLP), and the effectiveness is guaranteed by the inductive proof of the Bellman-Ford algorithm and the universal approximation theorem (Hornik, 1991).

Firstly, a well-learned MLP is expected to reconstruct $(k\text{-}1)$-hop limited shortest path distances from the inputs of $k$-th layer, with the existence of $\mathbb{A}'$-indicatable f-dist$^{k-1}$, which can be written as,

$$[dist_{\mathbb{A}'}^{k-1}]_{v_i} = \text{f-dist}_{\mathbb{A}'}^{k-1}(z_i^{k-1}) \tag{3}$$

then, considering the aggregate and update pattern of MPNNs, we expect an MPNN layer performed on node v to mimic a relaxation iteration on $v$. Deriving from the Bellman-Ford relaxation function (Equation 1), the expected function for an MPNN layer to mimic could be,

$$[dist_{\mathbb{A}'}^{k}]_{v} = \min(\{\min([dist_{\mathbb{A}'}^{k-1}]_{v}, [dist_{\mathbb{A}'}^{k-1}]_{u} + W_{e=\langle u,v \rangle}) | \forall u \in \mathcal{N}(v)\}) \tag{4}$$

or equivalently,

$$[dist_{\mathbb{A}'}^{k}]_{v} = \min([dist_{\mathbb{A}'}^{k-1}]_{v}, \min(\{[dist_{\mathbb{A}'}^{k-1}]_{u} + W_{e=\langle u,v \rangle} | \forall u \in \mathcal{N}(v)\})) \tag{5}$$

where the binary $\min$ function is expected to be mimicked by well-learned MLPs before pooling in the aggregator or the update function, while the $\min$ function over the set is expected to be mimicked by a well-learned parameterized sequential encoder like Recurrent Neural Networks (RNNs) or more commonly used max-pooling. A simple ordering of $\{dist_{\mathbb{A}'}^{k} | \forall \mathbb{A}' \in \mathbb{A}\}$ as part of output $Z^k$ would maintain the existence of $\mathbb{A}'$-indicatable f-dist$^k$.

There remains an issue of initialization. For the first layer, there is no natural existence of $\mathbb{A}'$-indicatable f-dist$^0$, as in the Bellman-Ford algorithm, the distance value of source nodes is initialized as 0, while others $\infty$. We would expect the updated distance value used in the $\min$ functions to be an enough large value, but it is not a natural way to explicitly assign it as node attributes, and we instead adopt a simple anchor indication strategy, or the f-ind with $\mathbb{A}'$-indicatability, where f-ind$_{\mathbb{A}'}(x_i)$ indicates whether the node $v_i$ is in $\mathbb{A}'$. And the expected function for the first MPNN layer to mimic, corresponding with equation 4 and 5, would be,

$$[dist_{\mathbb{A}'}^{1}]_{v} = \min(\{W_{e=\langle u,v \rangle} | u \in \mathcal{N}(v) \wedge \text{f-ind}_{\mathbb{A}'}(x_u)\}) \tag{6}$$

### 3.5 Problem Definition

The GIR framework is designed for node representation learning tasks, paying special attention to the capacity for getting position-aware embeddings. As a general encode for node representation learning, GIR takes graph $\mathbb{G} = (\mathbb{V}, \mathcal{E})$ with node attributes $X$ and optional edge weights $W$ as inputs, and embed nodes into $d$-dimensional vectors, represented as $Z \in \mathbb{R}^{|\mathbb{V}| \times d}$. The node representations are normally used in downstream tasks like node classification and link prediction.

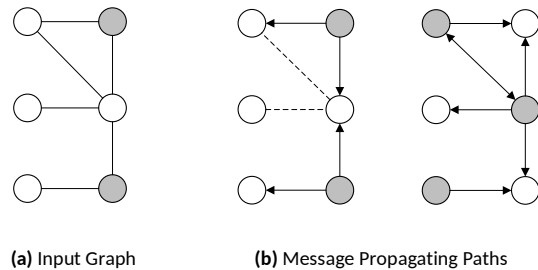

**(a)** Input Graph      **(b)** Message Propagating Paths

Figure 2: **(a)** Example input graph, colored nodes are chosen as anchors. **(b)** Message propagating edges for each layer in a 2-layer GIR model, Colored nodes highlight propagating source nodes.

## 4 METHODOLOGY

Here we proposed the GIR framework, acting as an extension of MPNNs to get position-aware embeddings. The GIR framework is based on the f-ind idea (Section 3.4) and a further specialization on the propagation paths, which we will introduce first, the overall design of the GIR framework will be elaborated then, finally, we take a discussion about using GIR as a general graph encoder.

### 4.1 GIR MODEL: MORE SPECIFIC PROPAGATION PATHS

Motivated by the relaxation order improved variant of the Bellman-Ford algorithm (Moore, 1959), we consider designing an MPNN architecture with more specific propagation paths. As the condition of whether to relax is dependent not only on the graph structure, we loose this condition to whether a node is reached by the breadth-first search from the source nodes. Under these considerations, we propose the Graph Inference Representation (GIR) model, which acts as a generalization of MPNNs, propagating messages from anchors along breath first search paths (Figure 2). The GIR model would be degraded to an MPNN or a non-graph encoder if selecting all nodes or no node for the anchor set.

There is a general consideration in MPNNs that how to deal with nodes without in-edge in the propagation step, and a common workaround is adding a self-loop to the input graph. In the GIR model, this becomes a more important issue as we additionally introduced this situation for nodes with the important role (anchor nodes that without other anchor nodes as a neighborhood). We take zero vector as a message if no in-neighborhood exists. In fact, considering a graph with only the same placeholders as node attributes and no node has zero in-degree, whatever the structure of the propagation tree is, all nodes are embedded identically. Adopting zero vector as dummy message is a simple but effective strategy for guaranteeing the existence of f-ind with $\{\mathbb{A}\}$-indicatability, thus a $k$-layer GIR with an anchor set $\mathbb{A}$ is $k$-hop $\{\mathbb{A}\}$-position-aware.

### 4.2 THE GIR FRAMEWORK

Having discussed the design choices for MPNNs and the propagating path variated GIR model, here we summarize the working process of the whole proposed framework for getting position-aware embeddings (Algorithm 1), building blocks are elaborated as follows.

**Anchor Selection** (Algorithm 1, line 1) The intuition behind anchor-based GNNs is that some nodes perform a more important role in the graph for the specific task (Liu et al., 2019; Xu et al., 2020). Specifing influential anchor node set is a highly data depent task, and it is a important research topic in the network analysis. We adopt degree centrality, the conceptually simplest choice, as a starting point. Also, the high interaction of degree centrality suggests an effective coverage through $k$-step message passing, and we heuristically believe it to be a promising choice for acting as positioning base. Concretely, we adopt degree centrality based GA-MPCA algorithm used in AGNN (Liu et al., 2019) as a default choice. For data or tasks with more heuristic, other choices may be adopted. For example, for web page networks, PageRank (Page et al., 1999) may be a more capable choice, and for networks constructed by snowball sampling (Goodman, 1961), seed nodes may be directly marked as anchor.

---

**Algorithm 1** The GIR framework.

---

**Input**: Graph $\mathcal{G} = (\mathcal{V}, \mathcal{E})$; Node input attributes $\{x_v, \forall v \in \mathbb{V}\}$; Edge weights $W$ (optional)
**Parameter**: Number of layers $L$; Anchor selection function $SelectAnchor$; Anchor indicatability mark function $IndicatableMark$; Message functions $M$; Neighborhood aggregator functions $AGGR$; Node update functions $U$.
**Output**: A-position-aware embedding $\{z_v, \forall v \in \mathbb{V}\}$

  1: $\mathbb{A} \leftarrow SelectAnchor(\mathcal{G})$
  2: $h_v^0 \leftarrow [x_v || IndicatableMark(v, \mathbb{A})], \forall v \in \mathbb{V}$
  3: $BFS \leftarrow \mathbb{A}$
  4: **for** $l \leftarrow 1$ **to** $L$ **do**
  5:   **for** $v$ in $\mathbb{V}$ **do**
  6:     $src \leftarrow$ a node set with lower bound $\mathcal{N}(v) \cap BFS$ and upper bound $\mathcal{N}(v)$
  7:     $m_v \leftarrow AGGR_{u \in src}^l M^l(h_u^{l-1}, h_v^{l-1}, W_{e=\langle u,v \rangle})$
  8:     $h_v^l \leftarrow U^l(h_v^{l-1}, m_v)$
  9:   **end for**
 10:   $BFS \leftarrow Successors(\mathcal{G}, BFS)$
 11: **end for**
 12: $z \leftarrow h^L$
 13: **return** $z$

---

**Anchor Indication**  (Algorithm 1, line 2) Assigning extend node attributes for guaranteeing the existence of f-ind with $\mathbb{A}'$-indicatability for all $\mathbb{A}' \in \mathscr{A}$ for a desired $\mathscr{A}$-position-aware capability. Here we introduce two simple strategies as the realization of f-inds, and note that those are optional for using the GIR model (Section 4.1) if only $\{\mathbb{A}\}$-position-aware capability is expected.

- **Anchor Labeling:** ($\{\mathbb{A}\}$-position-aware) Assigning one for anchor nodes as an extended attribute, and zero for others. It guarantees the existence of f-ind with $\mathbb{A}$-indicatability, by checking whether the extended attribute is one.

- **Anchor ID Labeling:** ($\mathbb{A}$-position-aware) Assigning one-hot labeling for anchors as extend attribute and zeros for others. For all anchor node $a \in \mathbb{A}$, the existence of f-ind with $\{a\}$-indicatability is guaranteed by indexing the ID of $a$ in the extended attributes. This strategy is similar to the node one-hot labeling trick but introducing much fewer input dimensions. Note that the anchor ID labeling is not capable of inductive settings, for it is sensible to the order of anchor nodes.

**Message Propagation**  (Algorithm 1, line 3–12) For the propagation paths, the GIR model (Section 4.1) gives a limitation for determining neighborhoods in the MPNN message-passing steps (tracked as $BFS$ in algorithm 1), and we take this limitation as lower bound in the whole framework. The message functions $M$ are preferred to be a parameterized encoder over node representations and optional edge weight, and the update functions $M$ are preferred to be a parameterized encoder over node message and representations, we commonly choose MLP as the parameterized encoders. Max-pooling is the most preferred choice for the aggregator $AGGR$.

## 4.3 GIR as General Graph Encoder

Although the starting point of GIR is to learn embeddings that are aware of position information measured by shortest path distance, design choices of GIR encourage it to be a general graph encoder with potential to be used for wider range of areas, here we discuss some related considerations.

Firstly, when using GIR model with a subset of nodes as anchor, information loss of graph structure would be occured. We suggest to apply GIR model on multiple anchor set and concatenate for outputs (named GIR-MIX), a mixture of GIR model and MPNN (GIR with all nodes as anchor) would be helpful for more general cases. Moreover, the mixture of models with $\mathscr{A}_1, \mathscr{A}_2, \cdots \mathscr{A}_n$-position-aware capability would be $(\cap_{i=1}^n \mathscr{A}_i)$-position-aware. Then for the issue of inductive setting, considering the components of GIR framework, common anchor selection methods with a general strategy are inductive, and the message passing is also inductive. And for the indication labeling strategy, only anchor labeling (which implies $\{\mathbb{A}\}$-position-aware) is inductive, order aware strategies like

Table 2: Results on synthetic datasets. **Bold font** highlights the best results.

| | $\{\mathbb{A}\}$-weighted | | $\{\mathbb{A}\}$-unweighted | | $\mathbb{A}$-weighted | | $\mathbb{A}$-unweighted | |
| --- | --- | --- | --- | --- | --- | --- | --- | --- |
| | mean | last | mean | last | mean | last | mean | last |
| MPNN | 3.4 | 3.6 | 0.6 | 0.7 | 4.0 | 4.0 | 0.5 | 0.5 |
| MPNN + anchor labeling | **0.1** | **0.1** | **0.0** | **0.0** | 3.5 | 3.5 | 0.5 | 0.5 |
| GIR | 0.2 | 0.2 | 0.0 | 0.0 | 3.5 | 3.5 | 0.5 | 0.5 |
|    w/ mean aggregator | 0.9 | 1.1 | 0.1 | 0.0 | 3.6 | 3.6 | 0.4 | 0.5 |
| MPNN + anchor ID labeling | — | — | — | — | 2.7 | 3.0 | 0.1 | 0.1 |
| GIR + anchor ID labeling | — | — | — | — | 2.5 | 2.7 | **0.1** | **0.1** |
|    w/ mean aggregator | — | — | — | — | 2.9 | 3.1 | 0.1 | 0.1 |
| GIR-MIX | — | — | — | — | **2.3** | **2.4** | 0.1 | 0.1 |

anchor ID labeling will be failed in inductive settings. If more position aware capability needed, particularly, GIR-MIX provide an inductive ability with pairwise tasks, which is same as in PGNN.

The time complexity of our default anchor selecting method is $O(|\mathbb{V}|^2)$ (Liu et al., 2019), and $O(|\mathcal{E}|)$ for commonly used adjacency list representation of graph. For the message propagation, we compute the total propagation edges, including message from self-loop. For a $K$ layer vanilla MPNN, the time complexity is $O(K|\mathcal{E}| + K|\mathbb{V}|)$, and for a $K$ layer GIR model, let $D$ represents the maximum of node out-degrees, we take the upper bound with $D$ that leads to $O(|\mathbb{A}|D^K + K|\mathbb{V}|)$, and it is strictly lower than or in the worst case equal to MPNN.

## 5 EXPERIMENTS

In this section, we first take experiments on a synthetic dataset to validate the effectiveness of the GIR framework for getting position-aware embeddings, then we demonstrate the performance of GIRs on real world datasets used in position-aware GNN literature.

**Datasets & Experimental Setup** Based on our position-aware definition (Definition 1), we construct four synthetic datasets, with $\{\mathbb{A}\}$/$\mathbb{A}$-position-aware information need on weighted/unweighted graphs, and following Velikovi et al. (2020), limited hop shortest path distances are predicted on each layer. We also take real-world datasets from previous position-aware GNN literature (You et al., 2019; Li et al., 2020). Those datasets with too few scales or in an inductive setting are filtered out. The graphs in these datasets are unattributed, the graph structure makes the major contribution to the tasks, and previous works have demonstrated the effectiveness of position information in those tasks, here we set node input attributes to ones as placeholders for a more fair comparison. More details of the datasets and eperimental setup are in Appendix A.1.

We choose baselines from MPNNs and anchor-based methods literature. The MPNN baseline uses a linear transformation for message and update functions, which keep the same as in our proposed models in the GIR framework. We take PGNN, AGNN and Distance Encoding (DE) as position aware GNN baselines, among those, PGNN and AGNN are anchor based methods, and the AGNN with fixed anchor is most related to our experiment setting. We take experiments for the GIR framework variants over: (1) MPNN or GIR message passing; (2) anchor labeling ($\{\mathbb{A}\}$-position-aware) or anchor ID labeling ($\mathbb{A}$-position-aware) anchorindication strategies; (3) max-pooling and mean aggregator; (4) GIR-MIX over all anchor node ($\mathbb{A}$-position-aware). Explicitly assigning distances and optional intermediate-hop distances in the anchor based setting are also experimented, and for synthetic datasets, those methods with explicit distance assigning are not evaluated.

**Results and analysis** For synthetic datasets, we report the mean absolute error (MAE) for mean outputs of every layer (mean) and on the last layer (last) in table 2. Considering that the average label value for these datasets (Table 4 in Appendix), the performance of the GIR framework with corresponding position-aware capability is quite promising. Models using mean aggregator are generally inferior to the corresponding max-pooling models, which supports max-pooling to be a preferred aggregator. Variants using GIR propagation get outperformed results in $\mathbb{A}$-position-aware datasets but inferior in $\{\mathbb{A}\}$-position-aware datasets. We owe the outperformed results to the

Table 3: Results on the position-aware real-world datasets, measured in test ROC AUC (in %) for link prediction (-lp) and node pair classification (-npc) task, and in test accuracy (in %) for node classification (-nc) task. ▲ highlights the improvement over the strong baseline DE, **Bold font** highlights top-3 results, * highlights the best results.

| | email-npc | europe-nc | usa-nc | ce-link | ns-link | pb-link |
|---|---|---|---|---|---|---|
| PGNN | 53.3 | 54.2 | 58.8 | 79.4 | **94.9** | 88.6 |
| AGNN | 89.6▲ | 52.4 | 61.4 | 84.9 | 89.7 | 93.4 |
| DE | 87.5 | **58.4** | 64.2 | 90.0 | **99.4*** | 95.0 |
| MPNN | 50.0 | 25.4 | 25.2 | 50.0 | 50.0 | 50.0 |
| w/ final distances | 99.7▲ | 50.0 | **72.4▲** | **91.5▲*** | 90.1 | 95.1▲ |
| w/ all distances | **99.8▲*** | **61.0▲*** | 71.5▲ | 87.7 | 88.2 | **95.4▲*** |
| MPNN + anchor labeling | 51.0 | 26.4 | 25.4 | 52.7 | 79.3 | 65.2 |
| GIR | 50.1 | 25.3 | 26.7 | 55.3 | 81.1 | 63.4 |
| MPNN + anchor ID labeling | **99.8▲*** | 52.2 | 66.0▲ | 89.6 | 85.9 | **95.3▲** |
| w/ final distances | **99.8▲*** | 50.6 | **72.5▲** | **91.3▲** | 90.0 | 95.2▲ |
| GIR + anchor ID labeling | **99.8▲*** | **56.7** | 67.6▲ | 89.9 | 89.9 | 95.1▲ |
| w/ final distances | 99.7▲ | 53.3 | **76.0▲*** | **91.3▲** | **91.2** | **95.3▲** |

more specific propagation paths in GIR models, thus more easily to learn the expected functions in the relatively complicated cases, while in the $\{\mathbb{A}\}$-position-aware datasets, both $\{\mathbb{A}\}$-position-aware models archives nearly perfect results, and the higher capacity of MPNN leads to a slight superiority.

Results of experiments on real-world datasets are reported in table 3, where the DE record the best performance of DE variants reported in the original paper (Li et al., 2020) for datasets except Email. We only keep part of the results for highlighting some key aspects, full results and more analysis are in Appendix A.3. Results show that with the same input feature, the MPNN baseline fails to distinguish different nodes. Explicitly assigning anchor based distances on MPNN improve performance by a large margin, and generally outperform or compete position-aware baselines, this demonstrates the effectiveness of the more flexible distance information interaction. Note that although accessing distance information to all anchor nodes, using this strategy alone is not $\mathbb{A}$-position-aware, as with no correspondence with anchor node ids. Anchor ID labeling provide slight improvement to this, and the GIR propagation strategy push the improvement further. And without explicit distance assigning, with $\{\mathbb{A}\}$-position-aware modeling ability, MPNNs with anchor labeling and vanilla GIRs generally get higher results, yet inferior to models with more powerful position-aware capability, and variants with GIR propagation perform generally better, these results suggest the effectiveness of the anchor indication and the further GIR propagation strategy.

In Europe-nc, the substantial improvement is got with all distance assigning, that suggests the effectiveness of intermediate limited hop distance information on the dataset, and in this case, GIR with anchor ID labeling outperforms those explicit final distance assigning variants, this suggest the potential advantage of the implicit learning strategy, as the information need by applying discrete graph algorithm may not clear in real-world scenarios. We ascribe the performance drop for assigning intermediate distance on other datasets to the information redundant. Moreover, we note that GIR framework performs inferior on Ns-link, and we ascribe it to the sparsity of NS dataset, unreachable nodes from anchors on the input graph will be not benefited from the GIR framework, and previous position-aware methods with special propagation strategy get relatively higher results.

## 6 CONCLUSION AND FUTURE WORK

We propose the GIR (Graph Inference Representation) framework, following the anchor-based GNN pattern and aims at conveying distance information implicitly along the MPNN message-passing steps. Theoretical implications and experimental results show the effectiveness of proposed strategies, and considerations for more general usages of GIR framework are discussed. Our empirical evaluations are taken on the synthetic datasets and relatively small scale real-world position-aware datasets used in the previous position-aware GNN literature, we leave the potential usage of GIR on larger scale scenarios for future work.

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

Table 4: Details of synthetic position-aware datasets.

| Dataset | $|\mathcal{G}|$ | $|\mathbb{V}|$ | avg. $|\mathcal{E}|$ | weighted | position-aware setting | avg. label | $|\mathbb{A}|$ | layers |
|---------|------|----------|-------|----------|------------------------|-----------|-----|--------|
| dist    | 200  | 100, 200 | 907.2 | ✓ | $\{\mathbb{A}\}$-position-aware | 8.1 | 5 | 3 |
| dist-u  | 200  | 100, 200 | 907.2 | ✗ | $\{\mathbb{A}\}$-position-aware | 1.6 | 5 | 3 |
| dist-a  | 1    | 2000     | 20000 | ✓ | $\mathbb{A}$-position-aware | 17.9 | 20 | 4 |
| dist-au | 1    | 2000     | 20000 | ✗ | $\mathbb{A}$-position-aware | 3.3 | 20 | 4 |

Table 5: Details of position-aware real-world datasets.

| Dataset | $|\mathbb{V}|$ | $|\mathcal{E}|$ | $|\mathbb{A}|$ | Cite |
|---------|------|-------|----|------|
| Email  | 1005 | 25571 | 50 | Leskovec et al. (2007) |
| Europe | 399  | 5995  | 20 | Ackland et al. (2005) |
| USA    | 1190 | 13599 | 50 | Ackland et al. (2005) |
| C.ele  | 297  | 2148  | 20 | Kaiser (2006) |
| NS     | 1461 | 2742  | 50 | Newman (2006) |
| PB     | 1222 | 16714 | 50 | Ackland et al. (2005) |

## A EXPERIMENT DETAILS

### A.1 DATASETS & EXPERIMENTAL SETUP

We first detail the construction of synthetic datasets. For $\{\mathbb{A}\}$-position-aware setting, we generate 160 graphs with 100 nodes each for training, 20/20 graphs with 200 nodes each as valid/test set, and for $\mathbb{A}$-position-aware setting, as anchor ID labeling strategy is not inductive, we generate one graph with 2000 nodes, train/valid/test sets are randomly split by 6:2:2. In the weighted setting, we assign edge weights as a uniformly sampled integer from 2 to 10, and in the unweighted setting, the distance between every connected node pair would be 1. Details of those synthetic datasets are listed in table 4.

For real-world datasets, following Li et al. (2020), we split train/valid/test sets by 8:1:1. Details of those real-world datasets are listed in table 5.

For each dataset, we report the test set performance at the best model with the valid set, and the final results are reported over 20 runs with different random seeds on 5 generated synthetic data samples or on the randomly split dataset for real-world datasets to keep consistent with DE baseline.

### A.2 IMPLEMENTATION DETAILS

Our implementation is based on Deep Graph Library (DGL) (Wang et al., 2019), with Py-Torch (Paszke et al., 2017) backend. For synthetic datasets, we use Adam optimizer for training, hyperparameters are set heuristically, listed in table 4. For real-world datasets, we mainly follow the hyperparameters settings in Li et al. (2020), with learning rate tuning in 1e-3 and 1e-4; hidden size set to 100; dropout set to 0.2; using 3 layer models for all datasets; the number of anchors are set heuristically according to the scale of each dataset and is also restricted by the GA-MPCA anchor selecting algorithm, listed in table 5.

### A.3 MORE RESULTS AND ANALYSIS

We note the full experiment results of real-world datasets (Table 6) and take some complementary analysis here.

We first note some key aspects for highlighting the improvements of GIR framework in table 7. We note the improvement of using explicit distance assigning on MPNN in (Table 7, 1-2), which highlights the effectiveness of distance information. Table 7, 3-4 lists the improvements of the GIR framework with different position-aware settings, settings with explicit distance assigning are not considered here for more clear comparison. The improvements of GIR propagation strategy are highlighted in table 7, 5-6, the comparisons are taken on MPNN with same position-aware

Table 6: Full results on the position-aware real-world datasets, measured in test ROC AUC (in %) for link prediction (-lp) and node pair classification (-npc) task, and in test accuracy (in %) for node classification (-nc) task. ▲ highlights the improvement over the strong baseline DE, **Bold font** highlights top-5 results, * highlights the best results.

| | | email-npc | europe-nc | usa-nc | ce-link | ns-link | pb-link |
|---|---|---|---|---|---|---|---|
| 1 | PGNN | 53.3 | 54.2 | 58.8 | 79.4 | **94.9** | 88.6 |
| 2 | AGNN | 89.6▲ | 52.4 | 61.4 | 84.9 | 89.7 | 93.4 |
| 3 | DE | 87.5 | 58.4 | 64.2 | 90.0 | **99.4*** | 95.0 |
| 4 | MPNN | 50.0 | 25.4 | 25.2 | 50.0 | 50.0 | 50.0 |
| 5 | w/ final distances | 99.7▲ | 50.0 | 72.4▲ | **91.5▲** | 90.1 | 95.1▲ |
| 6 | w/ all distances | **99.8▲** | **61.0▲** | 71.5▲ | 87.7 | 88.2 | **95.4▲** |
| 7 | MPNN + anchor labeling | 51.0 | 26.4 | 25.4 | 52.7 | 79.3 | 65.2 |
| 8 | w/ mean aggregator | 72.6 | 52.7 | 52.6 | 84.8 | 79.5 | 92.3 |
| 9 | w/ final distances | 99.7▲ | 50.3 | 72.2▲ | **91.6▲*** | 90.0 | 95.1▲ |
| 10 | w/ all distances | **99.8▲** | **61.0▲** | 70.5▲ | 87.6 | 87.9 | **95.5▲*** |
| 11 | GIR | 50.1 | 25.3 | 26.7 | 55.3 | 81.1 | 63.4 |
| 12 | w/ mean aggregator | 50.3 | 27.9 | 32.4 | 55.9 | 80.3 | 60.4 |
| 13 | w/ final distances | 99.7▲ | 52.2 | **74.7▲** | **91.3▲** | **91.2** | 95.3▲ |
| 14 | w/ all distances | **99.8▲** | 59.9▲ | 74.1▲ | 87.9 | 90.0 | **95.5▲*** |
| 15 | MPNN + anchor ID labeling | **99.8▲** | 52.2 | 66.0▲ | 89.6 | 85.9 | 95.3▲ |
| 16 | w/ mean aggregator | 99.6▲ | 51.4 | 66.1▲ | 89.0 | 67.0 | 94.7 |
| 17 | w/ final distances | **99.8▲** | 50.6 | **72.5▲** | **91.3▲** | 90.0 | 95.2▲ |
| 18 | w/ all distances | **99.8▲** | **62.1▲*** | 70.2▲ | 87.8 | 88.2 | **95.5▲*** |
| 19 | GIR + anchor ID labeling | **99.8▲** | 56.7 | 67.6▲ | 89.9 | 89.9 | 95.1▲ |
| 20 | w/ mean aggregator | **99.9▲** | 51.2 | 66.3▲ | 89.9 | 89.6 | 95.3▲ |
| 21 | w/ final distances | 99.7▲ | 53.3 | **76.0▲*** | **91.3▲** | **91.2** | 95.3▲ |
| 22 | w/ all distances | **99.8▲** | 59.2▲ | 74.7▲ | 88.1 | 90.0 | **95.5▲*** |
| 23 | MPNN + node one-hot labeling | **100.0▲*** | 48.5 | 61.3 | 87.9 | **91.8** | 93.7 |
| 24 | GIR + node one-hot labeling | **100.0▲*** | 59.5▲ | 68.6▲ | 88.4 | 86.6 | 94.1 |
| 25 | GIR-MIX | 65.6 | 27.9 | 35.4 | 81.3 | 86.3 | 84.6 |

Table 7: Highlights for key aspects of the results on real-world datasets (Table 6). Average improvement of the aspects on every datasets are noted (in %), references of comparisons are noted with the line number in table 6.

| | variants | references | email -npc | europe -nc | usa -nc | ce -link | ns -link | pb -link |
|---|---|---|---|---|---|---|---|---|
| 1 | MPNN w/ final distances | 5→4 | 99.4 | 96.8 | 187.3 | 83.0 | 80.2 | 90.2 |
| 2 | MPNN w/ all distances | 6→4 | 99.6 | 140.2 | 183.7 | 75.4 | 76.4 | 90.8 |
| 3 | {𝔸}-position-aware | 7→4,11→4 | 1.1 | 1.7 | 3.3 | 8.0 | 60.4 | 28.6 |
| 4 | 𝔸-position-aware | 15→4,19→4 | 99.6 | 114.4 | 165.1 | 79.5 | 75.8 | 90.4 |
| 5 | GIR (w/o distance assigning) | 11→7,19→15 | -0.9 | 2.2 | 3.8 | 2.6 | 3.5 | -1.5 |
| 6 | GIR (w/ distance assigning) | 13→9,14→10, 21→17,22→18 | -0.0 | 0.7 | 5.0 | 0.0 | 1.8 | 0.0 |
| 7 | w/ final distances | 17→15,21→19 | -0.0 | -4.5 | 11.1 | 1.7 | 3.1 | 0.0 |
| 8 | w/ all distances | 18→15,22→19 | 0.0 | 11.7 | 8.4 | -2.0 | 1.4 | 0.3 |
| 9 | 𝔸-position-aware over distance assigning | 17→5,18→6, 21→5,22→6 | 0.0 | 1.7 | 1.9 | 0.0 | 0.8 | 0.1 |

setting, settings of using explicit distance assigningare considered separately. The implicit encoding strategy, while has the capability of learning distances, due to the hardness of learning, assigning distance information explicitly would still be helpful. We note the improvement of explicit distance assigning over 𝔸-position-aware GIRs in table 7, 7-8. And the results show that the strategies for 𝔸-position-aware GIR would still be helpful with distance assigned (Table 7, 9).

For the choice of aggregator, we take max-pooling as the default following our starting point of mimicking Bellman-Ford algorithm, and the experiments on synthetic datasets support this, but it is not absolute for real-world cases as the information need may not clear. Results show that the mean aggregator (Table 6, 8,12,16,20) gets improvement on those $\{\mathbb{A}\}$-position-aware models (MPNN+anchor labeling and GIR), and for $\mathbb{A}$-position-aware models with anchor ID labeling, the need for more accurate distance information suggests the max-pooling aggregator.

The node one-hot labeling (Table 6, 23-24) provide a $\mathbb{V}$-position-aware capability to MPNN, besides the efficiency issue, results show a performance decline compared with using anchor ID labeling, this demonstrates the effectiveness of the important node selection strategy. Moreover, the GIR propagation with node one-hot labeling combine the merits of both, and gets a relatively promising result.

For GIR-MIX (Table 6, 25), with $\mathbb{A}$-position-aware, performs only superior to $\{\mathbb{A}\}$-position-aware models, and largely inferior to models with anchor ID labeling, we ascribe it to the extremely limited propagation path, though effective in the synthetic datasets, GIR-MIX for every single nodes may not be an effective general graph encoder.

