# OpenReview forum: "GIR Framework: Learning Graph Positional Embeddings with Anchor Indication and Path Encoding"
_ICLR.cc/2022/Conference — ICLR 2022 Submitted_

### Official Review · Reviewer_YkUE · 2021-10-26

**Correctness:** 3
**Technical Novelty And Significance:** 3
**Empirical Novelty And Significance:** 2
**Recommendation:** 5
**Confidence:** 3

**Main Review:**

Strengths:

1. The idea of mimicking Bellman-Ford algorithm sounds interesting and novel to the reviewer.

Weaknesses:

1. The presentation of the paper could be improved. The notation is a bit overwhelming, especially in Sec 3.3.  What's the difference of $\mathbb{B}$ and $\mathbb{N}$? What does $\mathbf{f}_{\mathbb{N}} (z) = f(z, N)$ mean? In addition, I suggest the authors to try to avoid using too many abbreviations in the experiment section as it's really hard to follow, e.g, Table 2, 4 (dist-u,a,au, -AA -A -M -O, etc.). Even the algorithm detail is not super clear to me. What does line 6 mean?

2. The experiments between MPNN and GIR in Table 2 is not consistent. I'm wondering why. It occurs to me that GIR has no significant benefts over MPNN.

3. The empirical study seems not sufficient. There's no large-scale experiments.

Questions:

1. Though the selection of anchors seem deterministic. I'm wondering what's the variance of selecting different anchors? Or what's the selecting of selecting 1 more or 1 less achors?

Post rebuttal

I thank the authors for the updates. The notation system looks better in the revised version, though I believe it is possible to deliver this paper in a clearer way. As for the empirical performance, I'm not convinced by the authors that MPNN-A is also a GIR framework (it's basically adding one hot labels to label nodes). Summarizing above, I believe this submission could be more impactful after careful revising. I will retain my original rating as borderline reject.

**Summary Of The Paper:**

This paper proposes the Graph Inference Representation (GIR) model, which first applies an anchor indication strategy and a forward process that mimics Bellman-Ford algorithm. The authors theoretically show that the proposed pipeline should be able to capture positional embeddings implicitly, and empirically show that the proposed pipeline can achieve superior performance on simulated and real-world datasets.

**Summary Of The Review:**

interesting idea, but the presentation is hard to follow and the empirical results do not seem very significant

---

> ### Author Response · Authors · 2021-11-17
> **Response to Reviewer YkUE (1/2)**
>
> We thank the reviewer for the valuable comments, here we address the concerns.
>
> > The presentation of the paper could be improved. The notation is a bit overwhelming, especially in Sec 3.3. What's the difference of $\mathbb{B}$ and $\mathbb{N}$? What does $f\_{\mathbb{N}}(z)=f(z,\mathbb{N})$ mean? In addition, I suggest the authors to try to avoid using too many abbreviations in the experiment section as it's really hard to follow, e.g, Table 2, 4 (dist-u,a,au, -AA -A -M -O, etc.). Even the algorithm detail is not super clear to me. What does line 6 mean?
>
> We are sorry for the confusing writing, notations, definitions, abbreviations and other writing issues are improved in the revised paper.
>
> For highlighting, $\mathscr{A}$($\mathbb{B}$ previously) is a set of node sets, and their union is the anchor set $\mathbb{A}$, we define position-aware on this for a high generality. While the $\mathbb{A}'$($\mathbb{N}$ previously) is a node set, we define indicatability on this as it is a more fine-grained constructing tool for leading to the position-aware embedding. And the $k$-hop $\mathscr{A}$-position-aware definition can be restated as the existence of $\operatorname{f-dist}$ with $\mathbb{A}'$-indicatability over $k$-hop limited multi-souce shorest path distance function, for all $\mathbb{A}'\in\mathscr{A}$.
>
> For $\operatorname{f}\_{\mathbb{A}'}(z\_i)=f(v\_i, \mathbb{A}')$(previously $\operatorname{f}\_{\mathbb{N}}(z)=f(z, \mathbb{N})$), $f$ is the function on a node and a node set, which is pre-defined and ready to be mimicked (multi-source shortest path distance function in our case), and $\operatorname{f}$ is a neural function, takes a embedding and output the mimicked result, the $\mathbb{A}'$ is not a argument of this function anymore.
>
> Please refer to the Section 3.2 \& 3.3 in the revised paper for more details.
>
> For the algorithm, we take the source nodes of GIR propagation as the lower bound, and MPNN propagation as upper bound, we have rewrite line 6 in the Algorithm 1 for more clarity.
>
> > The experiments between MPNN and GIR in Table 2 is not consistent. I'm wondering why. It occurs to me that GIR has no significant benefts over MPNN.
>
> We are sorry for the confusing representation. In fact, MPNN and GIR in the experiments represent different propagation strategies, with those anchor indication prefix, models like MPNN-A are also belong to the GIR framework.
>
> We have make a more clear organization of the experiments, please refer to Section 5 in the revised paper for more details.

---

> ### Author Response · Authors · 2021-11-17
> **Response to Reviewer YkUE (2/2)**
>
> > The empirical study seems not sufficient. There's no large-scale experiments.
>
> We first highlighting the main consideration and the contribution of our work. Our work is motivated by the position-aware GNN literature[2-4] and the neural Bellman-Ford[1]. We investigate the limitation about the information interaction of previous position-aware GNNs, and exploit more flexibility in the fixed anchor based setting. We propose a generalized neural Bellman-Ford with theoretically and empirically effectiveness to align anchor based setting, a specialized propagation strategy is further proposed. Experimental results demonstrates the effectiveness of our framework in the position aware scenarios.
>
> Indeed, we are aware of the usage of GIRs in more widely scenarios, and the design of GIR pays attention to this. We design GIR to be a general graph encoder, and that is why we focus on the implicit encoding strategy. The discussion of this consideration is indeed limited in the initial draft, and we have added more detailed discussion of more general GIR usages in Section 4.2 for anchor selection, and 4.3 for consideration for inductive setting and efficiency considerations.
>
> > Though the selection of anchors seem deterministic. I'm wondering what's the variance of selecting different anchors? Or what's the selecting of selecting 1 more or 1 less achors?
>
> Generally, anchors are preferred to be most important and influential nodes in the graph, and it is highly dependent on the task. The conceptually simplest degree centrality suggests an effective coverage through limited message passing, we heuristically believe it to be a promising choice for acting as positioning base, which is the starting point of position-aware GNNs. We have added more detailed discussion in the revised paper, Section 4.2.
>
> In fact, the datasets we taken in position-aware GNN literature is not very sensitive to this, just some positioning information is OK, with moderate anchor numbers, the effect of centrality measurements and slightly different anchor numbers is relatively minor.
>
>
> [1] Veličković, Petar, et al. "Neural execution of graph algorithms." arXiv preprint arXiv:1910.10593 (2019).
>
> [2] You, Jiaxuan, Rex Ying, and Jure Leskovec. "Position-aware graph neural networks." International Conference on Machine Learning. PMLR, 2019.
>
> [3] Liu, Chao, et al. "A-gnn: Anchors-aware graph neural networks for node embedding." International Conference on Heterogeneous Networking for Quality, Reliability, Security and Robustness. Springer, Cham, 2019.
>
> [4] Li, Pan, et al. "Distance encoding: Design provably more powerful neural networks for graph representation learning." arXiv preprint arXiv:2009.00142 (2020).

---

> ### Author Response · Authors · 2021-12-04
> **Further Clarification**
>
> We thank the reviewer for the reply, here we address the concerns.
>
> As presented in Section 4.2, the GIR framework is composed of message propagation paths with a lower bound as BFS from anchors, and the capability of indicating anchors. Though a simple strategy, the MPNN with anchor ID labeling (MPNN-A) is an instance of GIR framework with $\mathbb{A}$-position-aware capability, and as discussed in Section 3.4, it is capable to reconstruct the shortest path distances from every anchor nodes.
>
> For highlighting, our contribution is mainly centered on the clarified definition of position aware GNN on the fixed anchor setting, and we revisited the effectiveness of one-hot labeling strategy in this view.

---

### Official Review · Reviewer_C6uM · 2021-11-02

**Correctness:** 3
**Technical Novelty And Significance:** 3
**Empirical Novelty And Significance:** 3
**Recommendation:** 5
**Confidence:** 4

**Main Review:**

**Pros:**

- The idea of capturing positional information in the node embedding in an implicit way by constraining the propagation path in a way similar to BFS is interesting and could lead to powerful results.
- I appreciate the formalisation of positional-aware embeddings and the alignment between the proposed method and the BFS algorithm.

**Cons:**

- While the empirical results show some improvements in certain scenarios, it is not clear from the paper why or when the implicit positional encoding is superior to explicitly encoding of the shortest path. When the interactions between intermediate positional encoding (mentioned in the abstract) are useful, and why is it better to learn them instead of specifying them for each layer? A clear discussion in this regard would be useful to emphasize and clarify the theoretical contribution of this paper.
- Even if I find this idea as having big potential, I found the paper very hard to follow. The distinction between A-position-aware and {A}-position-aware is not clearly explained in the text even if it is a central element in the paper, as most of the experimental section refer to these two setups.
- The experiments presented in Table 5 are not enough explained in the text. What is the intuition for the difference in performance between GIR, GIR-A, GIR-O, and why is the ranking different from one dataset to another. What additional information brings each one of them? Moreover, I found the way the experiments are presented (too many abbreviations) difficult to follow.
- As the authors mentioned, the number of anchors and the way you pick them to influence the amount of structure captured by the algorithm, ranging from complete lack of structure to a normal message passing algorithm. Very few anchors or a bad way to choose them could lead to a great loss of information in terms of structure, eventually leading to poor results. An ablation study on how the results changed depending on this choice, or at least some intuition on what properties the SelectAnchor method should fulfil, would be useful.

Minors: The definitions for the concepts introduced are confusing, and some notations are not consistent along the paper. N is well-known for the set of natural numbers, so I recommend picking a different letter in definition 2. On page 7, there is a typo: I think it should be PGNN-F instead of PGNN-E; On page 7, Datasets paragraph, it should be Definition 1 instead of Definition 2; Same on the last paragraph before section 3.4; Multiple inconsistent usages of normal vs mathbb letters to denote sets (e.g. in Definition 2).

**Summary Of The Paper:**

The paper proposes a GNN method (GIR) that incorporates the paths produced by the BFS algorithm in the message passing algorithm, such that it enables the method to mimic the shortest path algorithm. This way, the method is able to learn positional embeddings implicitly. The experiments are conducted both on synthetic and real-world datasets.

**Summary Of The Review:**

Overall, I agree that the method could be useful and could improve the GNN methods in situations when the positional encoding is important. However, the current form of the paper lack clarity and I consider that many aspects of the method/experiments should be better present in order to be clearly delivered to the community.

---

> ### Author Response · Authors · 2021-11-17
> **Response to Reviewer C6uM**
>
> We thank the reviewer for the valuable comments, here we address the concerns.
>
> > it is not clear from the paper why or when the implicit positional encoding is superior to explicitly encoding of the shortest path. When the interactions between intermediate positional encoding (mentioned in the abstract) are useful, and why is it better to learn them instead of specifying them for each layer?
>
> Our claim of the advantage of implicit distance encoding strategy is based on the richness of interaction with distance information, the shortcoming of previous position aware GNNs in this view is mainly due to their model design. GIRs take the advantage of anchor based setting further and get improvement in this view, and the implicit encoding  retain potential possibility for more flexible usage of the graph data. We have revised the discussion about this in the abstract, Section 1, and Section 2.1.
>
> Reminded by your concerns, we have noticed that we lacked the experiments of explicit distance assign strategy based on the anchor based setting, this is not exist from previous works and gives a grateful insight to clarify our claim. We have added those experiments and update the result analysis in the revised paper.
>
> > The distinction between A-position-aware and {A}-position-aware is not clearly explained in the text
>
> We are sorry for the confusing writing, notations, definitions, abbreviations and other writing issues are improved in the revised paper.
>
> For highlighting, $\{\mathbb{A}\}$-position-aware embeddings capture the shortest path distance to any anchor node, and $\mathbb{A}$-position-aware to every anchor nodes. We have clarified this in the revised paper (Section 3.2).
>
> > What is the intuition for the difference in performance between GIR, GIR-A, GIR-O, and why is the ranking different from one dataset to another. What additional information brings each one of them?
>
> We have noticed that our previous experiment analysis lacked clarity. Having added the new experiment on the explicit anchor based distances assigning, the information need for datasets and the advantages of GIRs are delivered more clear. Please refer to the experiment section in the revise paper for more details.
>
> > As the authors mentioned, the number of anchors and the way you pick them to influence the amount of structure captured by the algorithm, ranging from complete lack of structure to a normal message passing algorithm. Very few anchors or a bad way to choose them could lead to a great loss of information in terms of structure, eventually leading to poor results. An ablation study on how the results changed depending on this choice, or at least some intuition on what properties the SelectAnchor method should fulfil, would be useful.
>
> Generally, anchors are preferred to be most important and influential nodes in the graph, and it is highly dependent on the task. The conceptually simplest degree centrality suggests an effective coverage through limited message passing, we heuristically believe it to be a promising choice for acting as positioning base, which is the starting point of position-aware GNNs.
>
> In fact, the datasets we taken in position-aware GNN literature is not very sensitive to this, just some positioning information is OK, with moderate anchor numbers, the effect of centrality measurements and slightly different anchor numbers is relatively minor.
>
> We highly agree that your concern is essential for more widely usage of GIRs, and we have added more detailed discussion about that in the revised paper. We have added the more detailed discussion of anchor selection in Section 4.2, and considerations for more general GIR usages in Section 4.3, the mixture strategy for information loss concern is discussed here.

---

> > ### Comment · Reviewer_C6uM · 2021-11-22
> > **Response to Authors' rebuttal**
> >
> > I would like to thank the authors for their rebuttal.
> >
> > After reading the rebuttal and the paper revision, I still have major concerns about the scenarios in which the proposed method is a better option than the existing ones.
> >
> > The results in Table 6 show a wide range of rankings between different positional-encoding choices. While I agree that this validates the fact that encoding positions in a GNN is a hard task, I still don’t fully understand what the cases when GIR offers advantages are. The proposed GIR represents the best option only in **3 out of 6 tasks**, and among these **2 represents tie results** between GIR and MPNN. I encourage the authors to try to emphasise their findings more. In this regard, a suggestion would be to also measure the average improvement of each important component presented in table 6, overall the datasets or any other way to extract useful information from this table. In the current form, I can only see that each problem has its own particular best architecture and I can’t actually identify what characteristic influence that.
> >
> >
> > I appreciate the effort made by the authors to improve the clarity of the paper. However, even in the current version, some elements are quite negligent. For example, in Table 3, it is mentioned in the caption that * denotes the best result but only 2 of the columns contains a *. In Table 6, some tie results for the best results are not both considered as best results (marked with *).

---

> > > ### Author Response · Authors · 2021-11-23
> > > **Response to Reviewer C6uM**
> > >
> > > We thank the reviewer for the feedback and the valueable suggestion, here we address the concern.
> > >
> > > Firstly, we are sorry for the confusing full result table (Table 6), there are too many factors for the ranking in one table. Reminded by your suggestion, we have highlights the key facts with the information extracted from table 6, please refer to the Appendix A.3 in the revised draft for more details.
> > >
> > > For highlighting, the GIR propagation strategy in fact shows improvement in 4 over 6 tasks and relativly minor decreases in the other 2 tasks (Table 7, 5). And in fact, as MPNN propagation has higher capacity compared with GIR propagation, with explicit distance assigned, the  advantage of GIR propagation is overlapped to some extent. That is a fatal factor to the overall ranking in table 6. More fair comparisons and highlights of our findings are noted in table 7.
> > >
> > > Also, as a side note, MPNNs with (anchor/anchor id) labeling are instances of GIR framework (Section 4.2).
> > >
> > > We are sorry for the negligence, mistakes are fixed in teh revised draft. And the best result mark in table 6 previously considered the exact value instead of the rounded one in the table.

---

### Official Review · Reviewer_Z48B · 2021-11-03

**Correctness:** 3
**Technical Novelty And Significance:** 2
**Empirical Novelty And Significance:** 2
**Recommendation:** 6
**Confidence:** 4

**Main Review:**

As the authors mentioned, the MPNN framework has inherent limitations, especially in position-aware embeddings.

So this work can bring some value to the GNNs community.

I have a question about GIR framework.

In Anchor Selection, it is determined in a heuristic way, degree centrality.

Because anchors are important components in the GIR framework, I believe there should be various attempts to choose the measure of importance for anchors.

In addition to degree centrality, Eigenvector Centrality, Katz Centrality, PageRank, Betweenness Centrality, Closeness Centrality, Harmony Centrality, and Clustering coefficient can be candidates for anchor selection.

Even some might be suited for effective coverage.

Or, additional trainable neural networks can be used. Could you explain how did you choose the measure and provide evidence as well?

**Summary Of The Paper:**

In this paper, the authors propose the anchor-based framework for position encoding.

It is well known that the message-passing framework inherently has a limitation to encode graph structure and sometimes fails to discriminate isomorphic subgraphs.

Graph Inference Representation exploits the distance relaxation process of the Bellman-Ford shortest path algorithm.

The authors show that the MPNN framework can keep track of the shortest paths using anchors.

To this end, the authors introduce the anchor labeling strategy for MPNNs mimicking the Bellman-Ford algorithm.

It has a more specialized structure and outperforms baseline models.

**Summary Of The Review:**

I would recommend this paper be accepted if other issues do not arise.

---

> ### Author Response · Authors · 2021-11-17
> **Response to Reviewer Z48B**
>
> We thank the review for the positive feedback and valuable comments, here we address the concerns.
>
> > In addition to degree centrality, Eigenvector Centrality, Katz Centrality, PageRank, Betweenness Centrality, Closeness Centrality, Harmony Centrality, and Clustering coefficient can be candidates for anchor selection.
> >
> > Even some might be suited for effective coverage.
> >
> > Or, additional trainable neural networks can be used. Could you explain how did you choose the measure and provide evidence as well?
>
> We highly agree your pointing out that the anchor selection strategy matters a lot, and we have taken a more detailed discussion in the revised paper (Section 4.2).
>
> For highlighting, anchors are preferred to be most important and influential nodes in the graph, and it is highly dependent on the task.  The conceptually simplest degree centrality suggests an effective coverage
> through limited message passing, we heuristically believe it to be a promising choice for acting
> as positioning base, which is the starting point of position-aware GNNs.
>
> In fact, we have taken some other choices in our early experiments, for these datasets empirically proven to need position information, with moderate anchor numbers, the effect of centrality measurements and slightly different anchor numbers is relatively minor.

---

### Official Review · Reviewer_jwSq · 2021-11-03

**Correctness:** 3
**Technical Novelty And Significance:** 2
**Empirical Novelty And Significance:** 2
**Recommendation:** 5
**Confidence:** 4

**Main Review:**

Strength:
This paper explores how to implicit learn graph position embeddings with anchor indication, and then propose a new strategy for message propagation under MPNN, which is clearly distinguished from other position-aware GNNs.

Weaknesses:

• The proposed framework (GIR) has limited theoretical contributions and  application scenarios, i.e., incapable of inductive settings, only tested on small-scale dataset. The empirical results lack the support for the key claim that the combined procedure of distance computing and encoding is more beneficial than the separated two-stage way.

• The authors only differentiate GIR with other position-aware GNNs, but do not present the connection and difference between previous works [1] (first propose max-pooling of MPNN to mimic Bellman-Ford Alg).

• The authors do not include the complexity discussion of the proposed method, especially regarding the part of labelling indication and
modified message passing and the cost comparison between implicit and explicit encodings. The introduction of relaxation process also raises the concern for the proposed method working on large-scale datasets.

• The notation system used in this paper appears to be very confusing. Several symbols are not properly defined, such as $f_p$, $g_a$ in Def 1.

[1] Veličković, Petar, et al. "Neural execution of graph algorithms." arXiv preprint
arXiv:1910.10593 (2019).


Typos:
In Def 1, $\mathbb{N} \in \mathbb{B}$ should be $\mathbb{N} \subset \mathbb{B}$?
In Def 2, $f_{\mathbb{N}}(z) = f(z, N)$ should be $f_{\mathbb{N}}(z) = f(z, \mathbb{N})$?
Reference for position-aware embeddings should be Def 1 instead of Def 2?

**Summary Of The Paper:**

This paper proposes a new type of anchor-based GNN by implicitly exploiting node positioning within customized message passing steps of MPNN. The framework consists of an anchor labeling strategy and a specified propagation path with the utilization of Bellman-Ford algorithm, to enable its positional awareness. The empirical results on small synthetic and real datasets show the effectiveness of implicit positional encoding, and achieve comparable results against other anchor-based models.

**Summary Of The Review:**

This paper presents an implicit way to learn graph positional embedding. However, the manuscript is not yet ready to be published. It does not provide a clear empirical investigation or a solid theoretical examination for how the implicit positional encoding is better than the explicit way. The content, organization and experiments also need to be further polished and enhanced.

---

> ### Author Response · Authors · 2021-11-17
> **Response to Reviewer jwSq**
>
> We thank the reviewer for the valuable comments, here we address the concerns.
>
> > The proposed framework (GIR) has limited theoretical contributions and application scenarios, i.e., incapable of inductive settings, only tested on small-scale dataset.
>
> Our work is motivated by the position-aware GNN literature[2-4] and the neural Bellman-Ford[1]. We investigate the limitation about the information interaction of previous position-aware GNNs, and exploit more flexibility in the fixed anchor based setting. We propose a generalized neural Bellman-Ford with theoretically and empirically effectiveness to align anchor based setting, a specialized propagation strategy is further proposed. Experimental results demonstrates the effectiveness of our framework in the position aware scenarios.
>
> In fact, the design of GIR pays attention to the more widely usages, and that is why we focus on the implicit encoding strategy and represent the method in a framework manner. We have added some considerations for more general GIR usages in Section 4.3, the suggested mixture strategy, inductive setting and efficiency issues are discussed.
>
> For highlighting the inductive setting issue, the anchor labeling strategy and GIR propagation are inductive, and more powerful strategy in the view of position-aware definition (Def 1) of taken the mixture of information propagated from all anchors has the same inductive ability with PGNN.
>
> > The empirical results lack the support for the key claim that the combined procedure of distance computing and encoding is more beneficial than the separated two-stage way.
>
> Our claim of the advantage of implicit distance encoding strategy is based on the richness of interaction with distance information, the shortcoming of previous position aware GNNs in this view is mainly due to their model design. GIRs take the advantage of anchor based setting further and get improvement in this view, and the implicit encoding  retain potential possibility for more flexible usage of the graph data. We have revised the discussion about this in the abstract, Section 1, and Section 2.1.
>
> Reminded by your concern, we have noticed that we lacked the experiments of explicit distance assign strategy based on the anchor based setting, this is not exist from previous works and gives a grateful insight to clarify our claim. We have added those experiments and update the result analysis in the revised paper.
>
> > The authors only differentiate GIR with other position-aware GNNs, but do not present the connection and difference between previous works [1] (first propose max-pooling of MPNN to mimic Bellman-Ford Alg).
>
> Our work is taken on a more generalized setting compared with [1], specific concern of the indicatability is taken, and we step further toward the specific propagation strategy design. We have detailed this in the revised paper (Section 2.2).
>
> > The authors do not include the complexity discussion of the proposed method, especially regarding the part of labelling indication and modified message passing and the cost comparison between implicit and explicit encodings. The introduction of relaxation process also raises the concern for the proposed method working on large-scale datasets.
>
> We thank the reviewer for pointing out this, and we have added this in Section 4.3. For the two building blocks of GIR framework, the anchor selection is generally efficient, for our default degree centrality the time complexity is $O(|\mathcal{E}|)$, and the GIR propagation is strictly more efficient than MPNN, and it is well supported by recent graph neural network engines (like DGL). As for the relaxation step, it is implicitly done in the message propagation steps.
>
>
> Also, we are sorry for the confusing writing, notations are improved and typos are fixed in the revised paper.
>
> [1] Veličković, Petar, et al. "Neural execution of graph algorithms." arXiv preprint arXiv:1910.10593 (2019).
>
> [2] You, Jiaxuan, Rex Ying, and Jure Leskovec. "Position-aware graph neural networks." International Conference on Machine Learning. PMLR, 2019.
>
> [3] Liu, Chao, et al. "A-gnn: Anchors-aware graph neural networks for node embedding." International Conference on Heterogeneous Networking for Quality, Reliability, Security and Robustness. Springer, Cham, 2019.
>
> [4] Li, Pan, et al. "Distance encoding: Design provably more powerful neural networks for graph representation learning." arXiv preprint arXiv:2009.00142 (2020).

---

### Author Response · Authors · 2021-11-17
**Summary of the First Revision**

We sincerely thank all the reviewers for their efforts and insightful comments. We have uploaded a new version of our draft, major revisions are summarized as follows.

- We make the claim of the advantages of GIR more clear with revisions in the abstract, end of Section 1, and Section 2.1.
- Add more details of the connection and difference between previous work for neural Bellman-Ford [1] at the end of Section 2.2.
- Improve the notations, and add more explanations of the introduced definitions in Section 3.2 \& 3.3.
- Add more details for the intuition of anchor selection in Section 4.2, "Anchor Selection".
- Add the discussion of the considerations for using GIR as a general graph encoder in Section 4.3, including the potential information loss consideration, inductive settings and the time complexity.
- Add variants that explicitly assign distance information in the experiment, and revise the result analysis to improve clarity. Confusing abbreviations are avoid.

[1] Veličković, Petar, et al. "Neural execution of graph algorithms." arXiv preprint arXiv:1910.10593 (2019).

---

### Comment · Area_Chair_4bti · 2021-11-20
**AC Discussion**

Thank you for responding to author feedback. We need more information to move forward. Could all of you read author rebuttals and other review comments to see if anything new to you? Please also raised further questions if you need more information to make a recommendation.
Thank you!
ICLR AC

---

### Author Response · Authors · 2021-11-23
**Summary of the Second Revision**

We sincerely thank the suggestion from Reviewer C6uM for improving clarity. We have highlighted the useful information extracted from the full result table (Table 6) in Table 7 and the disussion in Appendix A.3.

---

### Decision · Program_Chairs · 2022-01-20

**Decision:**

Reject

**Comment:**

This paper proposes to encode positions of nodes in graphs by anchor-based GNN with customized message passing steps. All reviewers raised significant concerns on this paper, including novelty of the message passing steps, experiments, writing and clarity, etc. The authors have actively responded to reviewer comments, but many of the concerns are still not addressed. Thus, the paper needs some work in order to be competitive.